# AMPK Activation Is Indispensable for the Protective Effects of Caloric Restriction on Left Ventricular Function in Postinfarct Myocardium

**DOI:** 10.3390/biology11030448

**Published:** 2022-03-16

**Authors:** Bernd Niemann, Ruping Pan, Hassan Issa, Andreas Simm, Rainer Schulz, Susanne Rohrbach

**Affiliations:** 1Department of Cardiac and Vascular Surgery, University Hospital Giessen and Marburg, Justus Liebig University Giessen, 35392 Giessen, Germany; bernd.niemann@chiru.med.uni-giessen.de; 2Department of Cardiothoracic Surgery, Martin Luther University Halle-Wittenberg, 06097 Halle, Germany; andreas.simm@medizin.uni-halle.de; 3Institute of Physiology, Justus Liebig University Giessen, 35392 Giessen, Germany; panruping1109@hotmail.com (R.P.); rainer.schulz@physiologie.med.uni-giessen.de (R.S.); 4Klinik für Kinder und Jugendliche, Evangelisches Krankenhaus Oberhausen, 46047 Oberhausen, Germany; hassan.issa@eko.de

**Keywords:** heart failure, ischemia, caloric restriction, AMPK, mitochondria

## Abstract

**Simple Summary:**

Impaired coronary blood flow induces cardiac ischemia and can lead to myocardial infarction. Although revascularization therapy and medical treatment potentially restore cardiac function in ischemic hearts, loss of cardiomyocytes and replacement of the dead myocardium by a scar can lead to ischemic heart disease with impaired cardiac output, a condition called heart failure. Medical treatment of heart failure includes drugs such as ACE-inhibitors, beta-blockers, diuretics, mineralocorticoid receptor antagonists, and SGLT (sodium-glucose cotransporter) 2 inhibitors. In addition, dietary interventions may provide further benefits to slow down disease progression. Caloric restriction, the decrease in calorie intake without malnutrition, is a strategy for improving health and increasing lifespan. Caloric restriction reduces the risk factors for cardiovascular diseases, and it improves heart function in animals and humans. Our study investigated the impact of caloric restriction on heart failure progression in mice and rats with myocardial infarction. We show that caloric restriction inhibits progressive loss of heart function, even when started after heart failure is already established. Together, the findings help to deepen our understanding of the complex mechanisms involved in the cardioprotective effects of caloric restriction and how they could be therapeutically utilized to prevent disease progression.

**Abstract:**

Background: Caloric restriction (CR) extends lifespan in many species, including mammals. CR is cardioprotective in senescent myocardium by correcting pre-existing mitochondrial dysfunction and apoptotic activation. Furthermore, it confers cardioprotection against acute ischemia-reperfusion injury. Here, we investigated the role of AMP-activated protein kinase (AMPK) in mediating the cardioprotective CR effects in failing, postinfarct myocardium. Methods: Ligation of the left coronary artery or sham operation was performed in rats and mice. Four weeks after surgery, left ventricular (LV) function was analyzed by echocardiography, and animals were assigned to different feeding groups (control diet or 40% CR, 8 weeks) as matched pairs. The role of AMPK was investigated with an AMPK inhibitor in rats or the use of alpha 2 AMPK knock-out mice. Results: CR resulted in a significant improvement in LV function, compared to postinfarct animals receiving control diet in both species. The improvement in LV function was accompanied by a reduction in serum BNP, decrease in LV proapoptotic activation, and increase in mitochondrial biogenesis in the LV. Inhibition or loss of AMPK prevented most of these changes. Conclusions: The failing, postischemic heart is protected from progressive loss of LV systolic function by CR. AMPK activation is indispensable for these protective effects.

## 1. Introduction

Age-associated diseases, including obesity, cardiovascular diseases (CVDs), and type 2 diabetes are rising, and there is an increasing need for efficient methods to tackle these diseases. Nutrition is a crucial determinant of aging. While chronic overnutrition leads to overweight, obesity, and type 2 diabetes, undernutrition can induce starvation effects, with an elevated risk of death. Dietary interventions in CVDs in patients do not primarily target lifespan but aim at delaying the onset or progression of age-related diseases. The sustained reduction in caloric intake, without malnutrition (caloric restriction, CR), is the most robust and best studied dietary intervention in numerous nonhuman species [1]. In addition, CR has also been shown to elicit systemic health benefits in humans who voluntarily or involuntarily adopt dietary restriction [2,3,4,5]. A recent meta-analysis of human randomized trials testing the cardiovascular effects of CR revealed a reduced cardiovascular risk, with significant blood pressure and heart rate lowering effects [6]. However, continuous CR has a poor long-term success rate, due to low adherence [7].

Many insights into the protective mechanisms of CR have been obtained from animal studies. Mild to moderate CR ameliorated cardiac dysfunction in different experimental settings. CR significantly reversed the aging phenotypes including cardiomyocyte hypertrophy and cardiac fibrosis, inflammation, and mitochondrial damage in middle-aged and old mice [8]. Lifelong CR has also improved age-related impairments in diastolic function in mice, as well as in Dahl salt-sensitive rats [9,10,11]. Previous studies from our group have shown that, even when started late in life, CR remained cardioprotective in senescent myocardium by correcting pre-existing mitochondrial dysfunction and apoptotic activation, an effect also depending on the duration and extent of CR [12,13,14]. Cardiac ischemic tolerance decreases with age, and many cardioprotective interventions, including ischemic preconditioning and postconditioning, are less effective in aged individuals [15,16]. Short- and long-term CR on the other hand conferred cardioprotection against ischemia/reperfusion injury in young and even in aged rodents [17,18,19]. CR causes a cascade of reactions at the levels of whole organism, organ, and individual cell to adapt to the energy limitation. The mechanisms underlying cardioprotective CR effects are diverse and include activation of SIRT1, AMP-activated kinase (AMPK), and peroxisome proliferator-activated receptor gamma coactivator 1-alpha (PGC-1alpha). AMPK is a key nutrient sensor, which is activated during starvation and maintains cellular energy balance. Activated AMPK phosphorylates key regulators of fatty acid, cholesterol, carbohydrate, and amino acid metabolism, as well as proteins involved in autophagy or cell growth [20]. In addition, AMPK regulates mitochondrial function through phosphorylation of proteins involved in mitochondrial biogenesis, mitochondrial fission, and mitophagy [20]. A strong activation of AMPK in the heart has been described after long- and short-term CR in mice and rats [13,17,19,21], but there are also reports suggesting that AMPK activation does not play a substantial role [22,23].

The present study investigated the role of AMPK activation in mediating the cardioprotective effects of CR on postischemic myocardium. We utilized a rat model of myocardial infarction (MI) and applied CR after heart failure development (4 weeks after MI), with or without application of the AMPK inhibitor compound C. In addition, the role of AMPK in mediating the CR effects in postischemic myocardium was also investigated in alpha 2 AMPK knock-out (KO) mice.

## 2. Materials and Methods

### 2.1. Rats and Diet Protocol

Male young (6 months) Sprague–Dawley rats were obtained from Charles River (Germany), caged individually with a light/dark cycle of 12 h and tap water ad libitum. Daily food intake of the normal standard diet was monitored for 14 days and averaged for each rat individually. Prior to surgery, animals received single subcutaneous (s.c.) injection of buprenorphine and 4 Vol% isoflurane until the rats reached a deep plane of anesthesia. The animals were intubated, ventilated with a 16-gauge intravenous catheter and placed in a supine position on a temperature control pad. After trimming the left side of the thorax with an electric razor and cleaning the surgical field with disinfectant, an anterolateral thoracotomy was performed under isoflurane anesthesia (2 Vol%) in fully ventilated rats. The incision between the third and the fourth rib was expanded by a retractor, and the pericardial sac was cut open to access the heart. After identification of the left anterior descending artery (LAD), a 6–0 silk ligature was passed underneath the LAD and tied with three knots. Visible blanching of the anterior wall of the left ventricle was indicative of successful ligation. Afterwards, the thorax was closed layer-by-layer. At the end of the surgery animals received s.c. buprenorphine, and the incision site was topically treated with bupivacaine. Sham-operated rats underwent the same procedure but without LAD ligation. 4 weeks after surgery, echocardiography was performed, and rats were assigned as matched pairs according to their LV systolic function to one of the following diets for the next 8 weeks (56 days). Rats on “control diet” (control) received their individual pre-diet average of Altromin^R^ 1244 (2550 cal/g), but not more, in order to avoid any degree of diet-induced obesity. Rats subjected to caloric restriction (CR) also received their pre-diet average, but of a calorically reduced, fiber-rich diet (Altromin^R^ 1344/1500; 1550 cal/g), which contains equivalent amounts of vitamins and minerals. In a second series of experiments, a subgroup of rats from both treatment groups (sham, infarct) received two intraperitoneal (i.p.) injections of the AMPK inhibitor compound C (Dorsomorphin 2HCl, Selleck Chemicals GmbH, # S7306, 10 mg/kg of body weight) during the last two weeks of the diet (day 43 and 50 of the diet). All animals were fasted for 12 h before sacrifice. At the end of the study, animals were anesthetized by isoflurane inhalation and euthanized by cervical dislocation in deep anesthesia. Blood was obtained by aortic puncture, and the LV scar was dissected from the non-infarcted myocardium. Animals were handled in accordance with a protocol approved by the animal care and use committee of the Martin Luther University Halle-Wittenberg (#2-784 MLU) and complied with the directive 2010/63/EU of the European Parliament on the protection of animals used for scientific purposes.

### 2.2. Mice and Diet Protocol

Mice with a targeted mutation of the alpha 2 AMPK (Prkaa2^tm1.1Vio^) [24] were purchased from Jackson Laboratory. Mice were housed in an air-conditioned room, with a 12 h light-dark cycle, and given standard chow with free access to tap water. Prior to surgery, wild-type (WT) and knock-out (KO) mice (male, 10–12 weeks old) received a single s.c. injection of buprenorphine and 4 Vol% isoflurane, until they reached a deep plane of anesthesia. Mice were intubated and ventilated with a Harvard Mouse Mini-Vent (Harvard Apparatus, March-Hugstetten, Germany), which supplied 0.2–0.25 mL room air 120 times per minute. After the chest was shaved and disinfected with 75% ethanol, a left thoracotomy was performed via the fourth intercostal space, and the lungs were retracted to expose the heart. After opening the pericardium, the left anterior descending coronary artery (LAD) was ligated with an 8–0 silk suture near its origin between the pulmonary outflow tract and the edge of the left atrium. Visible blanching of the anterior wall of the left ventricle was indicative of successful ligation. The lungs were inflated by increasing positive end-expiratory pressure, and the thorax was closed layer-by-layer. At the end of the surgery, animals received s.c. buprenorphine, and the incision site was topically treated with bupivacaine. Sham-operated rats underwent the same procedure but without any ligation. Four weeks after surgery, echocardiography was performed, and mice were assigned as matched pairs according to their LV systolic function and received control or CR diets, as described in rats, for the next 8 weeks. All animals were handled in accordance with a protocol approved by the animal care and use committee of the Martin Luther University Halle-Wittenberg (#2-666 MLU) and complied with the directive 2010/63/EU of the European Parliament on the protection of animals used for scientific purposes.

### 2.3. Echocardiography

Animals were anesthetized by isoflurane inhalation, and left ventricular function was assessed by two-dimensional echocardiography. Two-dimensional and M-mode echocardiographic examinations were performed in accordance with the criteria of the American Society of Echocardiography using a Vevo2100 system (FUJIFILM VisualSonics Inc., Toronto, ON, Canada), evaluating both cardiac geometry and function. They were conducted by the same trained, blinded sonographer. Left ventricular function was visually scanned by B-mode imaging in short and long parasternal axis. Measurement of left systolic and diastolic ventricular wall thicknesses and diameters, as well as measurement of aortal (Ao) and left atrial (LA) diameters, was performed in long parasternal axis by M-Mode imaging. Fractional shortening (FS) was calculated as FS = (LVEDD-LVESD/LVEDD) × 100, where LVEDD and LVESD are LV internal diameters in end-diastole and end-systole, respectively. Fractional shortening (% FS) and left ventricular ejection fraction (% EF) were calculated from mean values of six independently performed measurements per setting. Doppler tissue imaging from the apical four-chamber view was used to assess mitral valve annular velocities. Echocardiography of all animals was performed at the beginning of the diet (4 weeks after surgery) and at the end of the diet.

### 2.4. RNA Extraction

RNA was isolated from LV tissue with the RNeasy Mini Kit (Qiagen, Hilden, Germany, # 74106), according to the manufacturer’s instructions. Integrity and quality of the RNA was confirmed by agarose gel electrophoresis, and the concentration determined by measuring UV absorption.

### 2.5. Real-Time PCR

Reverse transcription of RNA samples (500 ng total RNA) was carried out for 30 min at 42 °C using the SuperScript™ III First-Strand cDNA Synthesis kit (ThermoFisher, Waltham, MA, USA, # 18080051). Real-time PCR (Primer sequences: Table 1) and data analysis were performed using the Mx3000P Multiplex Quantitative PCR system (Stratagene). Every reaction was performed as duplicates and quantified with the ΔΔC_T_-method. Threshold cycles (C_T_) of target genes were normalized to the mean of the housekeeping genes 18S rRNA, HPRT1, and GAPDH. The resulting ΔC_T_ values were compared to sham animals receiving control diet and relative mRNA expression was calculated by R = 2^−ΔΔCT^. For the estimation of mitochondrial DNA (mtDNA) content, DNA was extracted and purified from cardiac samples with the DNeasy Blood and Tissue kit (Qiagen). Relative copy numbers of mtDNA per diploid nuclear genome were measured using a fragment of mtDNA (16S rRNA), as well as a fragment of beta-globin. The relative fold change was then calculated using the ΔΔCT method.

### 2.6. Protein Extraction and Western Blotting

Frozen LV tissue was rapidly homogenized in a buffer containing 50 mM Tris HCl, 150 mM NaCl, 5 mM EDTA, 0.1% SDS, 1% sodiumdeoxycholate, and protease inhibitor cocktail (Sigma, St. Louis, MO, USA). Proteins were quantified using the BCA protein assay (Pierce). A total of 50 ¦Ìg of protein were loaded on a 10% SDS-PAGE gel. After electrophoresis, proteins were transferred to a nitrocellulose membrane. Filters were blocked and incubated with antibodies against cytochrome oxidase I (Cox I; ThermoFisher Scientific, Waltham, MA, USA, #459600), Cox 15 (abcam, #ab201082), Tfam (Santa Cruz, Santa Cruz, CA, USA, #sc-23588), PGC-1alpha (ThermoFisher, #PA5-72948), alpha AMPK (Cell Signaling, #2532), phospho-AMPK (Cell Signaling, #2535), VDAC1 (Cell Signaling, #4661), alpha 1 AMPK (abcam, Cambridge, UK, #ab32047), alpha 2 AMPK (abcam, #ab3760), Caspase-3 (Cell Signaling, Danvers, MA, USA, #14220 and #9664), and GAPDH (Cell Signaling, #2118). After incubation with a peroxidase-conjugated secondary antibody (1:10.000, Cell Signaling, #7074 and #7076), blots were subjected to the enhanced chemiluminescent detection method with the Fusion FX7 imaging system (Peqlab, Erlangen, Germany).

### 2.7. Respirometric Measurements

Immediately before oxygraphic measurements, LV fibers were permeabilized 30 min with saponin. After permeabilization the fibers were washed to remove saponin and adenine nucleotides. We used the high-resolution OROBOROS^®^ oxygraph, a two-chamber respirometer with a Peltier thermostat and integrated electromagnetic stirrers. Bundles of fibers (5–10 mg) were transferred into the oxygraph chambers. The measurements were performed at 30 °C in 1.42 mL incubation medium, consisting of 75 mM mannitol, 25 mM sucrose, 100 mM KCl, 10 mM KH_2_PO_4_, 0.5 mM EDTA, 5 mM MgCl_2_, 20 mM Tris-HCl, and 1 mg/mL BSA, (pH 7.4), using different substrates: 10 mM pyruvate + 2 mM malate and 10 mM succinate + 5 µM rotenone. The weight specific oxygen consumption was calculated as the time derivative of the oxygen concentration (DATGRAPH Analysis software, OROBOROS^®^). The rate of state 3 respiration was determined, following the addition of 5 mM ADP.

### 2.8. Measurement of LV ATP Content

ATP extraction from LV tissue was performed according to the protocol supplied by the manufacturer (ATP Colorimetric/Fluorometric Assay Kit, Sigma-Aldrich, St. Louis, MI, USA, # MAK190). LV tissue was removed from liquid nitrogen and immediately homogenized in 10 volumes of ATP assay buffer, followed by deproteinization with a 10 kDa molecular weight cut-off column (abcam, #ab93349). After incubation with reaction mix for 30 min, the fluorescence was measured (excitation 535 nm, emission 587 nm) on a Fluostar Optima microplate fluorometer (BMG Labtech, Ortenberg, Germany). An aliquot of non-deproteinized sample was utilized for determination of protein content using the BCA protein assay (Pierce).

### 2.9. Plasma Analyses

Plasma BNP, adiponectin, leptin and insulin concentrations were measured by using commercial enzyme-linked immunosorbent assays (rat BNP ELISA kit, Assay Pro, #ERB1202-1; mouse BNP ELISA kit, RayBiotech, Peachtree Corners, GA, USA, #EIAM-BNP-1; mouse/rat HMW adiponectin ELISA kit, BioVendor,#638-13079; mouse/rat leptin ELISA kit, BioVendor, Brno, Czech Republic, #RD291001200R; rat/mouse insulin ELISA kit, LINCO Research Inc., Saint Charles, MI, USA, #EZRMI-13K). Glucose concentrations were measured with the Glucose assay kit (BioCat, Heidelberg, Germany, #K686-100-BV). Triglycerides were measured using the GPO trinder kit (Sigma, #TR0100-1KT), and free fatty acids were measured with the Free Fatty Acid Quantitation kit (Sigma, #MAK044-1KT).

### 2.10. Statistical Analysis

Numeric parameters are expressed as mean ± SEM, unless otherwise stated. Data were analyzed for normal distribution (Shapiro-Wilk test) and variance (Levene test) and subsequently analyzed using two-way analysis of variance (all pairwise multiple comparison procedure; Holm Sidack’s test). Statistical significance was accepted at *p* < 0.05. We used SigmaStat for Windows version 3.5 (Systat Software Inc., Chicago, IL, USA). Sample sizes are provided in the figures.

## 3. Results

### 3.1. General Characteristics of the Rat Model

Four weeks after surgery, echocardiography was performed, and rats were assigned as matched pairs, according to their LV systolic function, to receive either the control or CR diets for the next 8 weeks. Body weight at the beginning of the diet did not differ between the groups (Figure 1A). CR resulted in a significant reduction in body weight and in LV weight normalized per tibia length in sham and infarct animals (Figure 1A). In addition, infarct-induced LV hypertrophy was diminished in CR infarct animals, compared to the infarct animals receiving the control diet, as demonstrated by changes in normalized LV weight (Figure 1A). In addition, we analyzed the expression of genes known to be part of the “hypertrophic gene program” [25]. Among these, the atrial natriuretic peptide (ANP) and beta-myosin heavy chain (beta-MHC) were significantly reduced in CR infarct animals, compared to infarct animals receiving the control diet (Appendix A). CR induced a strong increase in serum acrp30, a known physiological activator of AMPK, in sham animals (Figure 1B). In infarct rats, however, this CR-mediated effect was significantly blunted (Figure 1B). A comparable reduction in serum leptin occurred in sham and infarct animals after 8 weeks of CR (Figure 1B), an effect described before to occur with changes in body weight. In addition, fasting triglycerides were significantly reduced in both CR groups, compared to the according animals receiving control diet, while serum glucose or free fatty acids (FFA) were not altered (Figure 1B).

### 3.2. Characterization of Cardiac Function

LV systolic function, represented as LV fractional shortening (FS%), was significantly reduced 4 weeks after MI, demonstrating that the LV function of the infarct animals was significantly impaired before the dietary intervention. Eight weeks of control diet resulted in a further deterioration (Figure 2A). CR, on the other hand, induced a mild improvement in LV function, which was detectable when analyzing the changes in FS during the feeding period (Figure 2A). LV end-diastolic diameter (LVEDD) was significantly increased in infarct rats receiving the control diet, compared to CR rats at the end of the diet (without figure). Only a mild, non-significant reduction of heart rate was observed in response to CR in sham and infarct rats (not shown). The mRNA expression of brain natriuretic peptide (BNP), which is secreted by cardiomyocytes, in response to stretching, was strongly increased in the LV of infarct animals receiving control diet, compared to the according sham animals (Figure 2B). CR resulted in a significant reduction of BNP mRNA expression (Figure 2B). Similar results were also obtained for serum BNP (Figure 2B), suggesting that CR resulted in a reduced LV overload in rats with MI.

### 3.3. Signaling Pathway Activation and Mitochondrial Biogenesis

Activation of AMPK has been described to play a major role in the effects of long-term CR [13,17,19,21]. In addition to its role as a metabolic sensor in the heart, AMPK also influences other cellular processes, such as mitochondrial function, autophagy, mitophagy, oxidative stress, endoplasmic reticulum stress, and apoptosis [26,27]. Therefore, we compared AMPK activation, in response to 8 weeks CR or control diet, in cardiac tissue from sham and infarct rats. In addition, we analyzed parameters of apoptotic activation and mitochondrial biogenesis in these tissue samples. As shown in Figure 3A, sham animals showed a strong AMPK activation, in response to CR, while only a moderate increase in phosphorylated AMPK (pAMPK) was detectable in infarct rats after CR. Activation of caspase-3, an executioner caspase activated in the apoptotic cell by extrinsic (death ligand) and intrinsic (mitochondrial) pathways, was not detectable in cardiac tissue from sham animals (Figure 3B). Infarct animals demonstrated a mild caspase-3 activation, which was attenuated but not completely prevented by CR (Figure 3B). Among the markers of mitochondrial biogenesis, subunits of the cytochrome c oxidase (Cox I and 15) showed a marked increase in sham rats in response to CR, which was either blunted (Cox I) or completely absent (Cox 15) in infarct rats after CR (Figure 3C). Similarly, the protein expression of the mitochondrial porin VDAC1 showed a stronger induction after CR in sham rats, compared to infarct rats (Figure 3C). Interestingly, the protein expression of the two direct regulators of mitochondrial biogenesis, Tfam and PGC-1alpha, was not altered in response to CR (Figure 3C). Similar to the data obtained on the expression of Cox I and VDAC1 protein, mtDNA content was strongly increased in sham animals and mildly increased in infarct animals after CR, despite the lack of PGC1alpha and Tfam changes (Figure 3D).

### 3.4. Impact of AMPK Inhibition on CR-Induced Cardioprotection

In order to test the crucial role of AMPK on the observed effects in sham and infarct rats, we performed a second series of experiments, which included the application of a water-soluble inhibitor of AMPK (compound C), which has been previously used for in vivo studies [28,29]. Four weeks after surgery, rats were assigned as matched pairs, according to their LV systolic function, to control or CR diets for the next 8 weeks. A subgroup of rats received two i.p. injections of the AMPK inhibitor compound C during the last two weeks of diet. Changes in body weight did not differ between CR animals and those CR animals treated with compound C (not shown). As shown in Figure 4A, compound C did not alter LV systolic function in sham animals but prevented the CR-induced mild improvement in LV function in infarct rats. The CR-induced significant reduction in LV weight normalized per tibia length in sham and infarct animals was not altered by AMPK inhibition (Figure 4B). Accordingly, the infarct-induced LV hypertrophy remained diminished after compound C treatment in CR animals, compared to infarct animals receiving control diet (Figure 4B). However, the CR-induced reduction in LV BNP mRNA expression and BNP secretion into circulation was almost completely prevented by AMPK inhibition, compared to infarct rats receiving CR (Figure 4B). This suggests that AMPK plays indeed a crucial role in mediating the cardioprotective effects on LV function in rats with MI. No independent effect of compound C on cardiac function was observed in sham or infarct animals on control diet (Appendix A).

### 3.5. Impact of AMPK Inhibition on CR-Induced Effects on Signaling Pathway Activation, Mitochondrial Biogenesis and Respiration

As observed in the first cohort of rats, sham animals showed a strong AMPK activation in response to CR, while a slightly weaker increase in pAMPK was detectable in infarcts after CR. Compound C injection resulted in a significant attenuation of the CR effects on AMPK in both CR groups (Figure 5A). No independent effect of compound C on AMPK activation was observed in the sham or infarct animals on the control diet (Appendix A). The minor activation of caspase-3 (Figure 5B) detected in sham animals was strongly reduced by CR. This effect was completely reversed by AMPK inhibition (Figure 5B). Compared to sham, infarct animals demonstrated a stronger caspase-3 activation, which was attenuated by CR. Similar to the effects observed in sham animals, this effect was completely blocked by compound C (Figure 5B). Among the markers of mitochondrial biogenesis, Cox I, Cox 15, and VDAC1 showed a marked increase in sham rats in response to CR, which was also detectable, to a lesser extent, in infarcts rats (Figure 5C). Compound C totally abolished the effects of CR on Cox I, Cox 15, and VDAC1 protein expression in sham and in infarct rats (Figure 5C), but no independent effect of compound C on protein expression of Cox 15 and VDAC1 was observed in sham or infarct animals on the control diet (Appendix A). Tfam and PGC-1alpha protein expression was not altered in response to CR or compound C (Figure 5C).

In addition to changes in mitochondrial biogenesis, heart failure is known to induce an impairment in cardiac mitochondrial respiration. Permeabilized muscle fibers from LV tissue (Figure 6) showed a decreased pyruvate- and succinate-dependent state 3 respiration in infarct rats, compared to sham rats. CR and compound C treatment did not alter state 3 respiration in sham animals. In infarcts rats, however, CR induced a mild increase in pyruvate- and succinate-dependent state 3 respiration, which was completely abolished by AMPK inhibition (Figure 6).

### 3.6. Impact of Loss of Alpha 2 AMPK on CR-Induced Cardioprotection

Compound C has been suggested to mediate toxic cardiac effects, even at doses that do not alter AMPK activity [30]. In addition, compound C was reported to affect other signaling pathways [31]. Therefore, we decided to investigate the role of AMPK in CR in a second model. For this purpose, we utilized mice with a targeted mutation of alpha 2 AMPK [24], which is the dominant alpha AMPK isoform in muscle tissues. Furthermore, alpha 2 AMPK is crucial for protecting cardiac muscle from damage during ischemia [32]. As shown in the supplement, 8 weeks of CR induced a comparable 10–15% loss in body weight in sham and infarct mice, independent of their genotype (Appendix A). LV weight normalized per tibia length was significantly increased in WT and in KO mice following MI, and CR induced a comparable reduction in LV weight in both genotypes (Appendix A). However, the mRNA expression of the molecular markers of hypertrophy, ANP and beta-MHC, was only in WT CR mice significantly reduced, compared to WT infarct mice receiving control diet, while only a minor reduction in the mRNA expression of these genes in response to CR was observed in KO infarct mice (Appendix A). Comparable to the data obtained in rats, CR resulted in a strong increase in serum acrp30 in WT sham mice, but only in a moderate increase in WT infarct mice (Appendix A). AMPK KO mice, however, did not show any change in serum acrp30 after 8 weeks of CR (Appendix A). A comparable reduction in serum leptin and serum insulin occurred in sham and infarct mice, of both genotypes, after 8 weeks of CR (Appendix A). At the beginning of the diet, infarct mice of both genotypes demonstrated a significantly impaired LV function, compared to sham mice (Figure 7A). While 8 weeks of CR induced a mild improvement in LV function in WT infarct mice, compared to the according infarct mice receiving control diet, no comparable change in LV function was observed in KO mice (Figure 7A). In accordance with this, CR resulted in a significant reduction of BNP mRNA expression and BNP release in WT infarct mice, but not in KO infarct mice (Figure 7B).

### 3.7. Impact of Loss of Alpha 2 AMPK on CR-Induced Effects on Signaling Pathway Activation and Mitochondrial Biogenesis

CR induced a significant increase in AMPK phosphorylation in WT sham and infarct mice, while no activation of AMPK was observed in KO mice (Figure 8A). Loss of the muscle-specific alpha 2 AMPK resulted in a moderate upregulation of alpha 1 AMPK in KO mice (Appendix A). However, total alpha AMPK remained significantly lower in KO mice, compared to WT mice (Appendix A). MI did not cause an upregulation of alpha AMPK (Appendix A). In order to determine whether the loss of AMPK resulted in an accelerated ATP degradation, as described before, during exercise in alpha 2 AMPK-deficient muscles [33], we measured the tissue ATP content following CR. In WT mice, neither CR nor MI induced a drop in tissue ATP content (Appendix A). Loss of alpha 2 AMPK, on the other hand, was associated with a significant decrease in LV ATP content in sham animals on CR diet and in infarct animals receiving control or CR diets (Appendix A). The minor activation of caspase-3 (Figure 8B) detected in WT sham animals was strongly reduced by CR, an effect completely prevented by loss of alpha 2 AMPK (Figure 8B). In addition, sham KO mice showed stronger basal caspase-3 activation than sham WT mice (Figure 8B). Compared to sham, infarct animals demonstrated a stronger caspase-3 activation, which was attenuated by CR in WT mice and remained unaffected by CR in KO mice (Figure 8B). Among the markers of mitochondrial biogenesis, Cox I, Cox 15, and VDAC1 protein showed a marked increase in WT sham and infarct mice, in response to CR (Figure 8C). Loss of alpha 2 AMPK totally abolished the effects of CR on Cox 15 protein in sham and infarct KO mice, as well as on the Cox I and VDAC1 protein expression in infarct mice (Figure 8C). Sham KO mice experienced a mild increase in Cox I and VDAC1 protein expression, following 8 weeks of CR (Figure 8C). Tfam and PGC-1alpha protein expression was not significantly altered in response to CR, infarct or KO (Figure 8C).

## 4. Discussion

While the impact of CR on acute ischemia/reperfusion (I/R) injury has been extensively investigated [17,18,19,34,35,36,37,38] before, only few studies determined the potential protective effects of CR in postischemic, failing myocardium [39,40,41]. In a study by de Lucia et al. CR (intermittent fasting, IF) was initiated 4 weeks after MI and continued for 1 year [40], which is significantly longer than in our study. The authors reported that CR, when started after manifestation of heart failure, can ameliorate cardiac dysfunction, improve inotropic reserve, reduce cardiac fibrosis, and prevent β1-adrenoceptor downregulation in rats [40]. Others demonstrated that IF initiated after MI improves survival and reduced myocyte hypertrophy and LV dilation, although IF prior to LAD ligation did not acutely alter the MI size, compared to ad libitum fed rats [41]. In a further study, IF was initiated already two weeks after MI and lasted only 6 weeks [39]. This resulted in an improved long-term survival in animals with heart failure via proangiogenic, anti-apoptotic, and anti-remodeling effects [39]. Although all these analyses were performed in rats, the study protocols differ significantly, in terms of duration, initiation after MI, or feeding protocol. Nevertheless, all studies come to the unanimous conclusion that CR mediates cardioprotective effects in the postinfarct, failing myocardium. However, none of these studies was designed to investigate the exact role of one of the established mediators of cardiac CR effects. A major objective of our studies, performed in rats and mice, was the elucidation of the role of AMPK.

The mechanisms underlying the protective effects of CR are diverse, and many different mediators have been described. Among these mediators is AMPK. Although AMPK KO animals experienced a similar weight loss, many protective CR effects were abolished, verifying the crucial role of AMPK in mediating these effects. In addition, this suggests that weight loss per se is not sufficient to explain the cardioprotective CR effects. Changes in cellular adenosine monophosphate (AMP) or calcium (Ca2+) lead to the activation of AMPK via liver kinase B1 (LKB1) or Ca2+/calmodulin-activated protein kinase kinase-β (CAMKKß), respectively. In addition, AMPK can be activated by ischemia, hypoxia, hormones, oxidant signaling, cytokines, or adipokines [42]. Among these physiological mediators of AMPK activation are adipokines such as leptin [43], omentin [44], ghrelin [45], and acrp30 [46]. Acrp30 is secreted from adipocytes into circulation as high, medium, and low molecular weight (HMW, MMW, and LMW) forms. HMW has been shown to activate AMPK most potently in muscle cells [47]. The present study demonstrates an increase of serum acrp30 in the rat and mouse models, suggesting that the CR-induced increase in acrp30 may have contributed to AMPK activation. Interestingly, AMPK deficient mice did not show any increase in plasma acrp30 in response to CR. This suggests a direct impact of AMPK on acrp30 expression or release from adipose tissue. Conflicting results have been obtained regarding the stimulatory [48] or inhibitory effect [49] of AMPK activation on acrp30 expression. However, an increased multimerization of acrp30, leading to an increased release of HMW acrp30 from adipocytes, in response to the stimulatory effects of various AMPK activators, has consistently been described [48,49]. The ELISA kit utilized in our study detects HMW adiponectin, indeed suggesting an impact of AMPK on acrp30 multimerization. However, this remains speculative, since alpha 2 AMPK expression in adipose tissue is low compared to muscle tissues. We cannot exclude that the systemic effects of AMPK loss or pharmacological AMPK inhibition may have contributed to some of the cardiac changes observed in our animal models. A conditional alpha 2 AMPK KO should help answering this question. In addition, the mild effects of CR in KO sham mice, e.g., on markers of mitochondrial biogenesis, could also be mediated via alpha 1 AMPK, which shows a compensatory upregulation in KO animals, as observed in our study in the heart (Appendix A), but also by others in skeletal muscle [50].

The mouse model also shows that the postinfarct myocardium does not experience a generalized ATP deficiency. However, loss of AMPK, a crucial player in cellular energy homeostasis through activation of glucose and fatty acid uptake and oxidation in situations of low cellular energy supply, results in a drop in ATP content after CR or MI. Accordingly, mice with a genetic long-term inhibition of AMPK activity demonstrate an impaired recovery of LV contractile function during postischemic reperfusion that was associated with a lower ATP content and increased apoptotic activation [51]. In addition, the role of AMPK as a guardian of mitochondrial homeostasis, though regulation of mitochondrial biogenesis and mitophagy, may have contributed to the reduced ATP content. This reduction in ATP has been reported to occur in animals after MI [52], but also in hearts from AMPK KO mice under CR [21], by others before. It is also in accordance with previous reports showing that muscle-specific loss of alphas 1 and 2 AMPK results in an ATP depletion during treadmill exercise [53], suggesting that AMPK is necessary for maintaining the cellular nucleotide pool in situations with increased energy demand. Although AMPK is a key regulator of mitochondrial biogenesis through activation of PGC-1alpha and nuclear respiratory factor-1 (NRF-1) [54,55,56], no change in PGC-1alpha protein expression was observed in our rat or mouse models. Unlike this, a reduced expression of PGC-1alpha and Tfam has been demonstrated in hypertrophied and failing hearts in many studies [57]. However, some studies found no alterations in PGC-1alpha expression [58,59,60]. Indeed, PGC-alpha dysregulation occurs not only at the level of mRNA or protein expression but through epigenetic, post-transcriptional, and post-translational regulatory mechanisms [57,61]. Thus, an unaltered PGC-1alpha expression, as observed in our study, does not exclude an altered PGC-alpha activity and, thus, an impaired expression of PGC-alpha target genes.

Dysfunctional mitochondria providing insufficient ATP for essential cellular functions may also have contributed to the increased apoptotic susceptibility of the postinfarct myocardium. In addition, AMPK was shown to mediate anti-oxidative and anti-apoptotic effects in cardiomyocytes or hearts [51,62,63]. Currently, a number of questions with major impact towards a translation of dietary interventions into clinical practice remain unanswered, including the overall impact of CR on quality of life, hospitalization, and mortality. Furthermore, whether the beneficial effects of CR persist in aged animals, or animals with various comorbidities, or whether the benefits of CR are preserved after diet discontinuation has not yet been investigated. Various drugs activate AMPK including the insulin-sensitizing biguanide metformin, statins, and thiazolidinediones. Considering the potential side effects of chronic CR and the difficulties with long-term adherence to such a dietary regimen in patients, the clinical application of established drugs capable of AMPK activation may provide a supplemental potential to treat patients after MI, in order to preserve LV function [42].

Our study has a few limitations. Only male animals were utilized in the present study, limiting a general translation into clinical situations. There are well-documented sex-dependent differences in lifespan extension and health benefits induced by CR in rhesus monkeys, rodents, C. elegans, and humans [64,65,66,67,68], with mostly stronger and more consistent effects in males. The prevalence of ischemic heart disease (IHD) is lower in females than in males [69], and many mechanisms involved in ischemic heart disease or the response to myocardial ischemia reperfusion injury differ between sexes [70,71]. Although the higher incidence and prevalence of IHD may justify an initial analysis of CR-mediated cardioprotection in male animals, the data obtained in our study are primarily meaningful only for males, while their transmissibility to females is uncertain. Thus, more studies allowing a direct side-by-side comparison of the effect of CR on males and females of the same species are desirable in the future. In addition, we cannot estimate exactly how differences in activity levels between animals on control diet or CR, as demonstrated by our group previously [14], have contributed to the cardioprotective effects of CR. Indeed, it has been shown that a combination of CR and exercise exerts greater cardioprotection than a monotherapy in female insulin resistant rats [72]. However, no additive effects on cardiovascular disease risk factors were observed in humans, when weight loss was matched [73].

## 5. Conclusions

The present study reports that CR, started when HF is already established, inhibits progressive loss of LV systolic function and attenuates LV remodeling. In addition, our study shows in two independent animal models and two different species in which the AMP-dependent protein kinase plays a crucial role in the cardioprotective CR effects.

## Figures and Tables

**Figure 1 biology-11-00448-f001:**
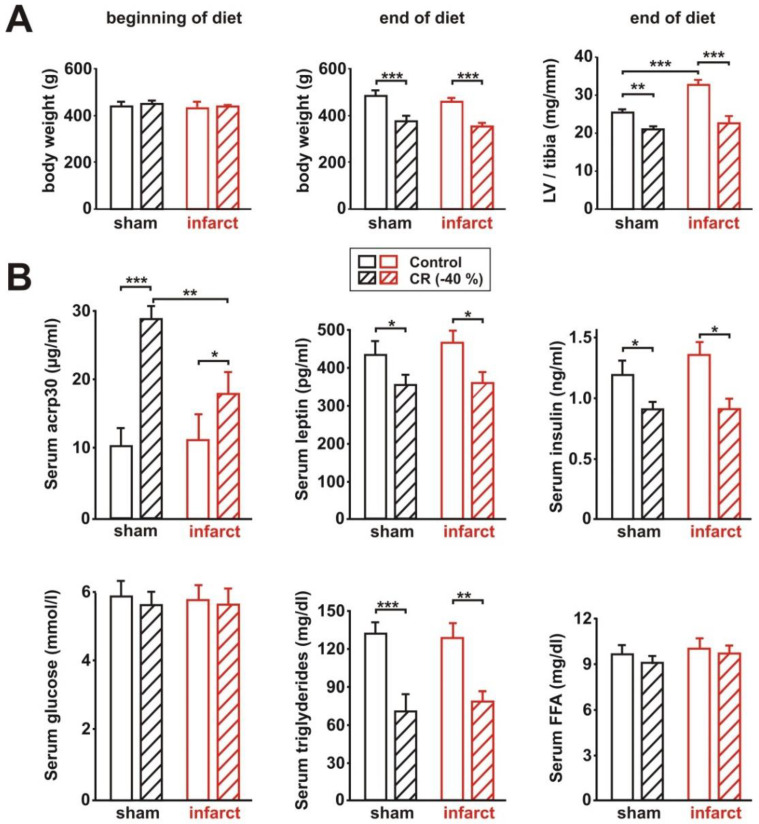
Effect of 8 weeks CR on body weight and plasma parameters. (**A**): Body weight at the beginning of the diet (left panel), end of the diet (middle panel), and LV weight normalized to tibia length at the end of the diet (right panel) in sham or infarct rats, following 8 weeks of control or CR diets. (**B**): Serum levels of acrp30, leptin, insulin, glucose, triglycerides, and free fatty acids (FFA) in these rats, obtained after overnight fasting. All data are mean ± SEM. *n* = 5–6 animals per group, *: *p* < 0.05; **: *p* < 0.01; ***: *p* < 0.001.

**Figure 2 biology-11-00448-f002:**
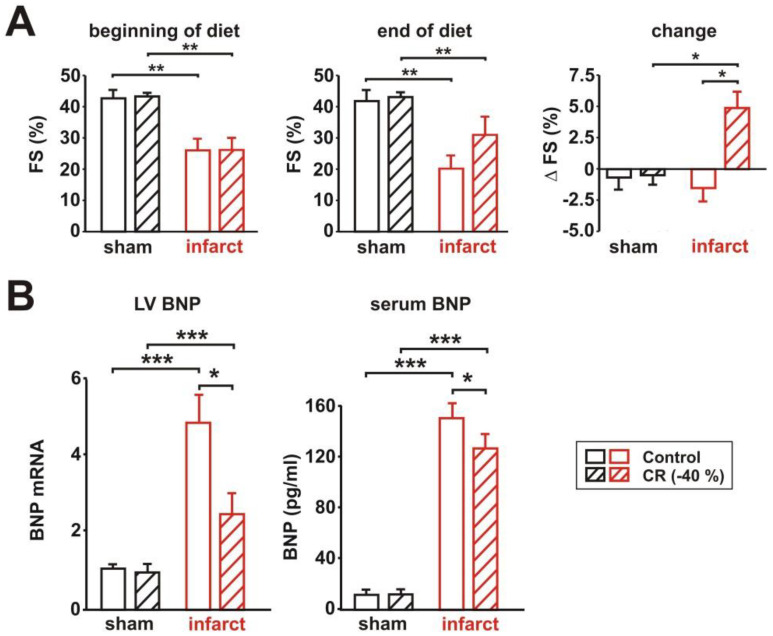
Effect of 8 weeks CR on LV systolic function and expression or release of brain natriuretic peptide (BNP). (**A**): LV fractional shortening (FS%) at the beginning of the diet (left panel), end of the diet (middle panel), and changes in FS during the feeding period (right panel) in sham or infarct rats following 8 weeks of control or CR diets. (**B**): LV mRNA expression (left panel) and serum levels of brain natriuretic peptide (right panel) in sham or infarct rats, following 8 weeks of control or CR diets. All data are mean ± SEM. *n* = 5–6 animals per group, *: *p* < 0.05; **: *p* < 0.01; ***: *p* < 0.001.

**Figure 3 biology-11-00448-f003:**
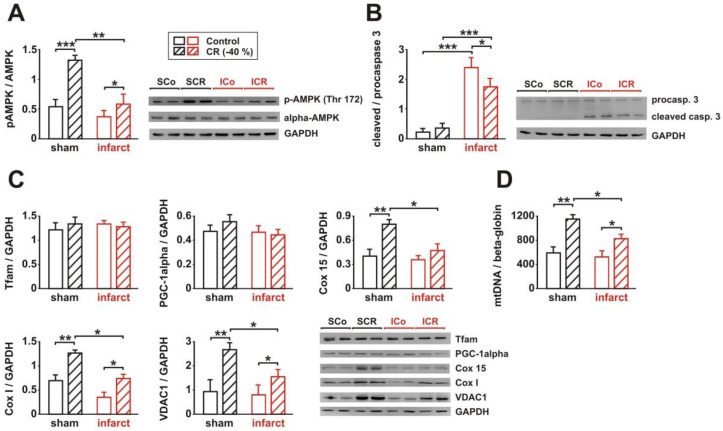
Effect of 8 weeks CR on AMPK activation, apoptosis, and mitochondrial biogenesis. (**A**): Representative Western blots and quantification of phosphorylation of AMPK (at Thr172), normalized to alpha AMPK, in the LV of sham or infarct rats, following 8 weeks of control or CR diets. (**B**): Representative Western blots and quantification of full-length and large fragment (17/19 kDa) of cleaved, activated caspase-3 in the LV of sham or infarct rats, following 8 weeks of control or CR diets. GAPDH served as loading control. (**C**): Representative Western blots and quantification of protein expression of Tfam, PGC-1alpha, mitochondrial components of complex IV (Cox 15 and I) and VDAC1, normalized to GAPDH, in LV tissue. (**D**): Relative copy number of mtDNA per diploid nuclear genome, as measured by real-time PCR. All data are mean ± SEM. *n* = 5–6 animals per group, *: *p* < 0.05; **: *p* < 0.01; ***: *p* < 0.001. Full size blots are provided in Appendix A.

**Figure 4 biology-11-00448-f004:**
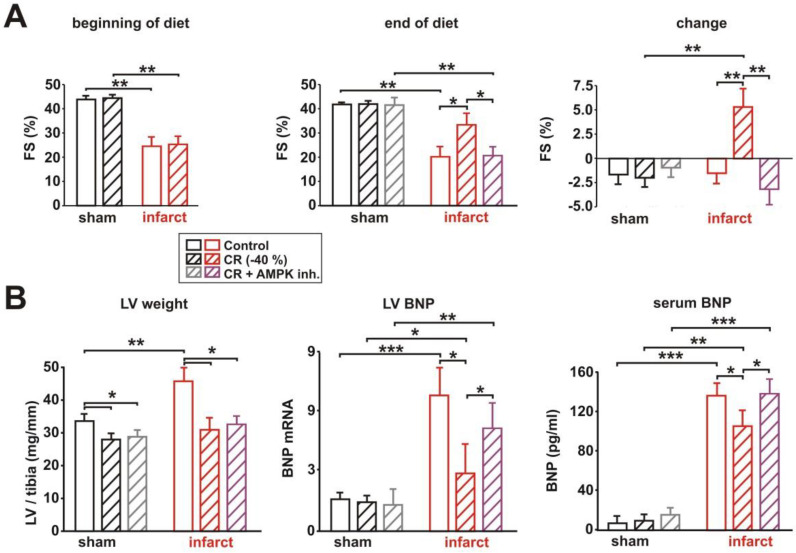
Effect of 8 weeks CR and 2 weeks of AMPK inhibition on LV systolic function, LV weight and expression or release of BNP. (**A**): LV fractional shortening (FS%) at the beginning of the diet (left panel), end of the diet (middle panel), and changes in FS during the feeding period (right panel) in sham or infarct rats, following 8 weeks of control or CR diets. (**B**): LV weight normalized to tibia length at the end of the study (left panel), LV mRNA expression (middle panel), and serum levels of brain natriuretic peptide (right panel) in sham or infarct rats following 8 weeks of control or CR diets. All data are mean ± SEM. *n* = 6–7 animals per group, *: *p* < 0.05; **: *p* < 0.01; ***: *p* < 0.001.

**Figure 5 biology-11-00448-f005:**
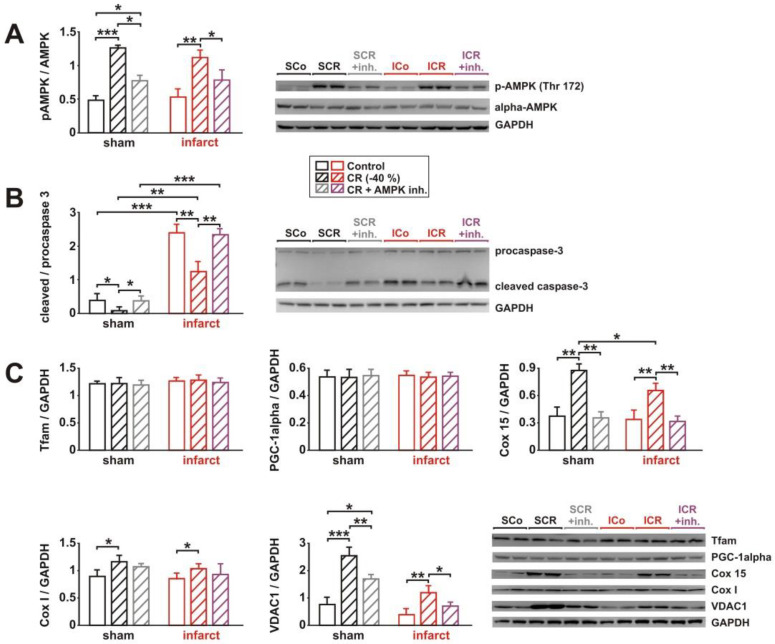
Effect of 8 weeks CR and 2 weeks of AMPK inhibition on AMPK activation, apoptosis, and mitochondrial biogenesis. (**A**): Representative Western blots and quantification of phosphorylation of AMPK (at Thr172), normalized to alpha AMPK, in the LV of sham or infarct rats, following 8 weeks of control or CR diets and AMPK inhibition. (**B**): Representative Western blots and quantification of full-length and large fragment (17/19 kDa) of cleaved, activated caspase-3 in the LV of sham or infarct rats, following 8 weeks of control or CR diets and AMPK inhibition. GAPDH served as loading control. (**C**): Representative Western blots and quantification of protein expression of Tfam, PGC-1alpha, mitochondrial components of complex IV (Cox 15 and I), and VDAC1, normalized to GAPDH, in LV tissue. All data are mean ± SEM. *n* = 6–7 animals per group, *: *p* < 0.05; **: *p* < 0.01; ***: *p* < 0.001. Full size blots are provided in Appendix A.

**Figure 6 biology-11-00448-f006:**
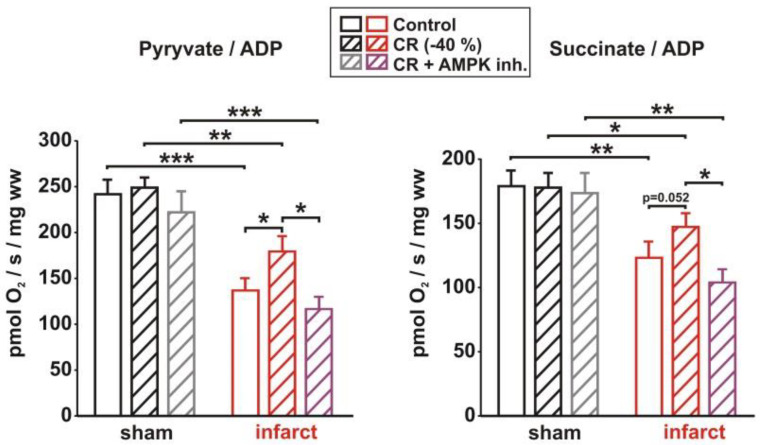
Effect of 8 weeks CR and 2 weeks of AMPK inhibition on mitochondrial respiration. Active rates of respiration (state 3) were measured in saponin-skinned cardiac fibers in the presence of 5 mM ADP and either 10 mM pyruvate + 2 mM malate or 10 mM succinate in the presence of 5 µM rotenone in the LV of sham or infarct rats, following 8 weeks of control or CR diets and AMPK inhibition. All data are mean ± SEM. *n* = 5 animals per group, *: *p* < 0.05; **: *p* < 0.01; ***: *p* < 0.001.

**Figure 7 biology-11-00448-f007:**
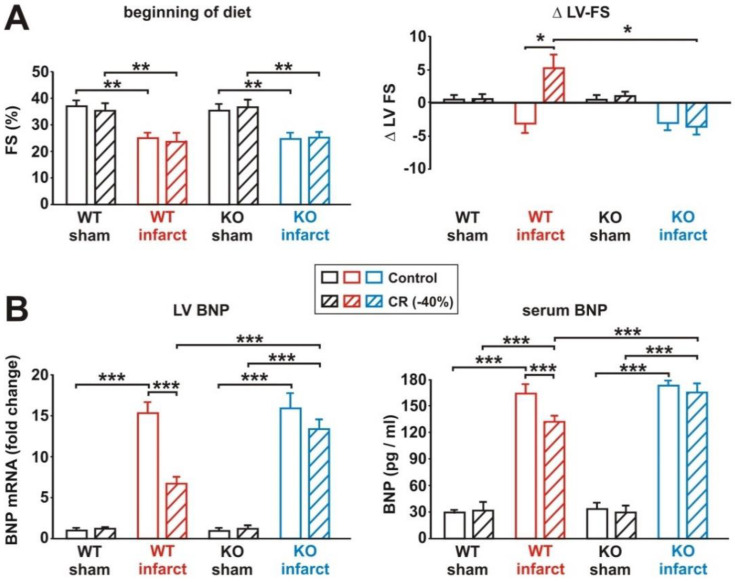
Effect of 8 weeks CR on LV systolic function and expression or release of BNP. (**A**): LV fractional shortening (FS%) at the beginning of the diet (left panel) and changes in FS during the feeding period (right panel) in WT and KO sham or infarct mice, following 8 weeks of control or CR diets. (**B**): LV mRNA expression (left panel) and serum levels of brain natriuretic peptide (right panel) in WT and KO sham or infarct mice, following 8 weeks of control or CR diets. All data are mean ± SEM. *n* = 6–8 animals per group, *: *p* < 0.05; **: *p* < 0.01; ***: *p* < 0.001.

**Figure 8 biology-11-00448-f008:**
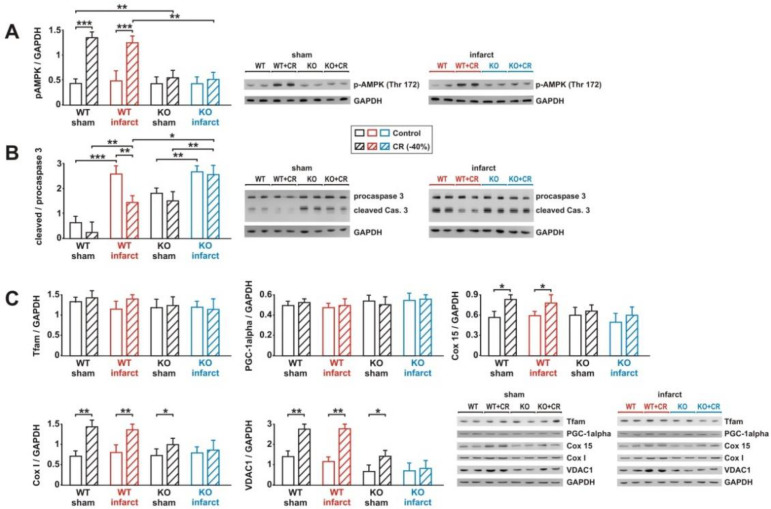
Effect of 8 weeks CR on AMPK activation, apoptosis, and mitochondrial biogenesis. (**A**): Representative Western blots and quantification of phosphorylation of AMPK (at Thr172), normalized to GAPDH, in WT and KO sham or infarct mice, following 8 weeks of control or CR diets. (**B**): Representative Western blots and quantification of full-length and cleaved, activated caspase-3. GAPDH served as loading control. (**C**): Representative Western blots and quantification of protein expression of Tfam, PGC-1alpha, mitochondrial components of complex IV (Cox 15 and I), and VDAC1, normalized to GAPDH, in LV tissue. All data are mean ± SEM. *n* = 6–8 animals per group, *: *p* < 0.05; **: *p* < 0.01; ***: *p* < 0.001. Full size blots are provided in Appendix A.

**Table 1 biology-11-00448-t001:** Primer sequences.

Primer	Accession Number	Sense	Antisense
BNP rat	NM_031545	CTCAAAGGACCAAGGCCC TAC	CTGCCCAAAGCAGCTTGAAC
BNP mouse	NM_008726	CTGAAGGTGCTGTCCCAGAT	CCTTGGTCCTTCAAGAGCTG
ANP mouse	NM_008725	ATGGGCTCCTTCTCCATC	GTGTTGGACACCGCACTGTA
ANP rat	NM_012612.2	GAGCGAGCAGACCGATGAA	GATCTATCGGAGGGGTCCCA
beta-MHC mouse	NM_080728.2	GAGAAGATGTGCCGGACCTT	GGACAGCTCCCCATTCTCTG
beta-MHC rat	NM_017240.2	TGCTCTACAATCTCAAGGAGAGGT	AGGCGTTGTCAGAGATGGAGA
mt. DNA rat	KM577634.1	AGTGAAGGGGCGGATCATA	GAGGTCACCCCAACCGAAAT
beta-globin rat	NM_001113223.1	GCCTGTGGGGAAAGGTGAATG	CTTCACCTGGGGGTTACCCAT
HPRT rat	NM_012583.2	ACCAGTCAACGGGGGACATA	ATTTTGGGGCTGTACTGCTTGA
HPRT mouse	NM_013556.2	GATCAGTCAACGGGGGACAT	AGAGGTCCTTTTCACCAGCAA
GAPDH rat	NM_017008.4	CACCATCTTCCAGGAGCGAG	GAAGGGGCGGAGATGATGAC
GAPDH mouse	BC023196.2	CATCACCATCTTCCAGGAGCG	CGTTTGGCTCCACCCTTCAA
18S rRNA	NR_046237	TGGAGCGATTTGTCTGGTTA	ACGCCACTTGTCCCTCTAAG

## Data Availability

The data presented in this study are available on request from the corresponding author.

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
