# Peer review of "AMPK Activation Is Indispensable for the Protective Effects of Caloric Restriction on Left Ventricular Function in Postinfarct Myocardium"

_biology, 2022, doi:10.3390/biology11030448_

Round 1

Reviewer 1 Report

This study describes a set of experiments and a set of studies looking at the effects of caloric restriction on measures of cardiovascular health. The authors demonstrate how caloric restriction affects a number of different measures of cardiovascular health and function and shows a role for some of the signaling pathways.  I only found one typo on line 124:  Serious should be series

Author Response

We have corrected this mistake accordingly.

Reviewer 2 Report

Niemann et al. conducted a study looking at the cardioprotective role of CR in a murine model of MI by LAD ligation. The authors show that the cardioprotective effect of CR is driven by enhancing AMPK activity under stress conditions; therefore reflecting an improvement in mitochondrial function and a decrease in myocardial apoptosis. Here are my concerns/suggestions that may improve the quality of the work:

  • In figure 1, the authors showed a mild improvement in FS in MI CR vs MI no calorie restriction. Where all echocardiographic data among groups obtained at similar heart rates? It would be important to clarify.
  • VDAC1 expression is usually unaltered between normal and heart failure condition. Is there any change in mt-content between CR vs no calorie restriction groups?
  • It is clearly established in the literature that PGC-1a and TFAM are downregulated in HF vs Sham. Moreover, PGC-1a transcriptional regulation and function is regulated by AMPK. In the current study, both PGC-1a and TFAM expression was unaltered among the different group. The authors should clarify to why their findings differ from previous work.
  • Minor: Line 124, please change “serious” to “series”. Line 128, please change “killing” to “Sacrifice”.  

Reviewer 3 Report

This study Niemann et al examines the ability of post-ischemic cardiac dysfunction to be mitigated by caloric restriction that is initiated after the pathological stressor. As the authors note in the Discussion, a protective effect of this intervention is not a new discovery, but the investigation of a possible mechanism and the data implicating AMPK is. The design of the study answers the questions posed and the various approaches to determine mechanism strengthen the claim of AMPK involvement. Overall, this is an interesting study with some novel findings that are of interest to the broader research community.

That being said, I do have some issues that require attention.

  1. The Introduction is excessively long. Focusing on a few studies to establish the benefits of caloric restriction, along with a more focused development of the AMPK hypothesis would be helpful.

  1. Why are only male rats used? There needs to be a robust, scientific justification for the focus on a single sex. The authors note that this is a limitation, but this is not a sufficient justification for choosing males instead of females, or running a sex comparison. Given the acknowledged differences in caloric restriction studies this is a significant concern.

  1. The timing of the AMPK inhibitor injections needs clarification. What days were the injections done? "During the last two weeks" is insufficient.

  1. It's unclear why caloric restriction was done in rats, but the molecular mechanisms determined in mice. While the justification for genetically engineered mice is understandable, why not do the caloric restriction protocol in mice to allow for a more direct comparison?

  1. The text and Figure 1A (second panel) claim to show reduced body weight, but the first panel of Figure 1A appears to show no differences. This is confusing and could be rectified by simply plotting weights before and after infarct and caloric restriction.

  1. The claim is made that hypertrophy is diminished in caloric restricted infarct rats, but no true measurements of hypertrophy were done. There is a decrease in LV-tibia length, but this could be atrophy by cell loss and not a decrease in hypertrophy or cell size per se. Measurements of cardiomyocyte size by histology or markers of hypertrophy need to be completed to substantiate this claim.

  1. Many of the figures involving the mice show the same data repeatedly. For example, the data in Figure 4 are the same as those presented in Figure 2. These should be presented once as one figure. Similarly, Figures 3 and 5 as well as 7 and 8 have repeated data that should be presented once. While the combination of these data sets will reduce the number of figures, the amount of data is more than sufficient for a full study.

  1. The statement in the Discussion that plasma acrp suggests that this was a driving factor in AMPK activation should be tempered to so it "could" be a factor. (line 482).

Round 2

Reviewer 2 Report

The authors have addressed all the raised concerns. I have no additional comments at this time.